# Endogenous and Exogenous Modulation of Nrf2 Mediated Oxidative Stress Response in Bovine Granulosa Cells: Potential Implication for Ovarian Function

**DOI:** 10.3390/ijms20071635

**Published:** 2019-04-02

**Authors:** Omar Khadrawy, Samuel Gebremedhn, Dessie Salilew-Wondim, Mohamed Omar Taqi, Christiane Neuhoff, Ernst Tholen, Michael Hoelker, Karl Schellander, Dawit Tesfaye

**Affiliations:** 1Institute of Animal Science, Department of Animal Breeding and Husbandry, University of Bonn, 53175 Bonn, Germany; okha@itw.uni-bonn.de (O.K.); seta@itw.uni-bonn.de (S.G.); dsal@itw.uni-bonn.de (D.S.-W.); mtaq@itw.uni-bonn.de (M.O.T.); cneu@itw.uni-bonn.de (C.N.); etho@itw.uni-bonn.de (E.T.); mhoe@itw.uni-bonn.de (M.H.); ksch@itw.uni-bonn.de (K.S.); 2Teaching and Research Station Frankenforst, Faculty of Agriculture, University of Bonn, 53639 Königswinter, Germany; 3Center of Integrated Dairy Research, University of Bonn, 53175 Bonn, Germany

**Keywords:** Nrf2 signaling pathway, miRNAs, oxidative stress, quercetin, bovine granulosa cells

## Abstract

Nrf2 is a redox sensitive transcription factor regulating the expression of antioxidant genes as defense mechanism against various stressors. The aim of this study is to investigate the potential role of noncoding miRNAs as endogenous and quercetin as exogenous regulators of Nrf2 pathway in bovine granulosa cells. For this cultured granulosa cells were used for modulation of miRNAs (miR-28, 153 and miR-708) targeting the bovine Nrf2 and supplementation of quercentin to investigate the regulatory mechanisms of the Nrf2 antioxidant system. Moreover, cultured cells were treated with hydrogen peroxide to induce oxidative stress in those cells. Our results showed that, oxidative stress activated the expression of Nrf2 as a defense mechanism, while suppressing the expression of those miRNAs. Overexpression of those miRNAs resulted in downregulation of Nrf2 expression resulted in higher ROS accumulation, reduced mitochondrial activity and cellular proliferation. Quercetin supplementation showed its protective role against oxidative stress induced by H_2_O_2_ by inducing the expression of antioxidant enzymes. In conclusion, this study highlighted the involvement of miR-153, miR-28 and miR-708 in regulatory network of Nrf2 mediated antioxidant system in bovine granulosa cells function. Furthermore, quercetin at a low dose played a protective role in bovine granulosa cells against oxidative stress damage.

## 1. Introduction

In mammals, the female reproductive performance is controlled by complex regulatory mechanisms in ovarian tissues in order to develop a competent oocyte, which can induce pregnancy following fertilization and early embryo development [1]. During the reproductive lifespan, a cow is exposed to various stresses, including metabolic and heat stress, which results in disturbance of the redox hemostasis and development of oxidative stress [2,3,4]. Oxidative stress is a phenomenon when the amount of reactive oxygen species (ROS) surpasses the cellular ability to counteract via generating scavenging antioxidant response system [5]. ROS are reactive chemical species containing various forms of oxygen such as superoxide anion (O_2_•^−^), hydrogen peroxide (H_2_O_2_), and hydroxyl radical (HO^•^) [6]. ROS affects cellular functions by promoting cellular proliferation, differentiation, autophagy and apoptosis [7,8]. Various studies have revealed the crucial role of ROS as central elements in cell signaling [9,10] and gene expression [11]. Furthermore, ROS play a crucial role in follicular development [12], oocyte maturation [13], fertilization and embryo development [14]. However, over accumulation of ROS resulted in hazardous effect on cellular micro- and macromolecules such as DNA, proteins and lipids, which are associated with cellular death [15]. Thus, maintaining the ROS at the basal level is essential for normal cellular functions. Recent study on the outcome of polycystic ovary syndrome (PCOS) in human on oocyte quality reported that, higher intracellular ROS level in PCOS granulosa cells induced cellular apoptosis, which contributes to poor oocyte quality and reduced the positive pregnancy outcomes [16]. Thus, maintaining the cellular homeostasis via maintaining the equilibrium between ROS and cellular antioxidant machinery is required for the oocyte development and competency [17].

The nuclear factor (erythroid-derived 2)-like 2 (*NFE2L2* or *Nrf2*) is a redox sensitive transcription factor that regulates various genes involved in a various cellular functions and protection against endogenous and environmental stressors [18,19,20,21]. Under normal conditions, *Nrf2* is kept at basal homeostatic level in cytosol by its inhibitory protein; *Keap1* [22]. Upon exposure to stressors’ stimuli, *Nrf2* is dissociated from *Keap1* and subsequently is translocated into the nucleus, binding to an antioxidant response element (ARE) located at upstream promoter region of its downstream antioxidant target genes such as superoxide dismutase (*SOD1*), NAD(P)H:quinone oxidoreductase (*NQO1*), Peroxiredoxin (*PRDX1*) and catalase (*CAT*) [23,24,25,26] and facilitate the scavenging of the excessive ROS. In our previous studies, we have demonstrated the emerging role of *Nrf2*/*Keap1* signaling pathway under oxidative stress conditions in bovine preimplantation embryos [27] and granulosa cells [28]. In addition to the Keap1-mediated posttranslational control of *Nrf2*, another level of regulatory network of *Nrf2* through transcriptional, translational and other posttranslational modifications are reported [21]. Recently, it was shown that the *Nrf2* signaling pathway could be regulated epigenetically via DNA methylation, histone modifications and interactions of microRNAs (miRNAs) [29].

MicroRNAs are short non-coding RNAs of 18–24 nucleotides long that play crucial role in posttranscriptional endogenous gene regulators, by binding to the 3′-untranslated region (3′-UTR) of the target mRNA resulting in either degradation of the target mRNA and\or reduced translation [30,31]. Species and tissue expression patterns of miRNAs have been reported previously [32,33,34,35]. Our previous studies showed the crucial role of miRNAs in several biological processes in bovine granulosa cells including proliferation, differentiation, stress response and apoptosis [36,37,38,39]. Other studies have also evidenced the involvement of miRNAs in the regulation of reproductive functions including folliculogenesis [40,41,42], oocyte maturation [43,44], corpus luteum function [45,46] and early embryonic development [47,48,49]. *Nrf2* has been reported to be regulated by several miRNAs including miR-28 in breast cancer cells MCF-7 and embryonic kidney cells 293T (HEK293T) [50] and miR-153 in breast cancer cell lines [51], neuroblastoma cells [52,53], glioma stem cells [54] and primary hippocampal neurons [55]. However, so far the regulatory role of those miRNAs (miR-28, miR-153 and miR-708) in bovine granulosa cells especially in response to exposure to oxidative stress has not been investigated. 

Several attempts have been carried out to use antioxidant supplementation as a means of counteracting the oxidative stress conditions by restoring cellular antioxidant defense mechanism. For instance, flavonoids, which are natural phytochemical compounds, exert antioxidant effects against oxidative stress conditions [56]. Among flavonoids, quercetin (Que) (2-(3,4-dihydroxyphenyl)-3,5,7-trihydroxy-2,3-dihydrochromen-4-one) considered the strongest antioxidant, which inhibit oxidant damage via different pathways [57,58,59]. Previous studies showed that, Que has a broad range of pharmacological properties including antioxidant effect [60,61,62,63], anti-inflammatory effect [64,65,66] and anti-apoptotic effects [63]. Furthermore, Que is considered as a candidate chemo-preventive against oxidative stress in different human cell types [67]. Previous studies indicated that, Que modulates several signaling pathways mainly the *Nrf2* signaling pathway and interacts with cellular antioxidants defense system such as NAD(P)H:quinone oxidoreductase (*NQO1*), glutathione S-transferase (*GSTs*), thioredoxin (*TRX*) and heme oxygenase 1 (*HO-1*) in rodents and humans [68,69,70]. However, the exogenous regulation of *Nrf2* signaling pathway via Que in bovine granulosa cells and subsequent cellular functions is still unknown.

Therefore, our study was designed to investigate the endogenous regulatory mechanisms of the *Nrf2* signaling pathway via miRNAs and exogenous factor Que in bovine granulosa cells functions. Here, we specifically focused on the potential involvement of candidate miRNAs namely; miR-153, miR-28 and miR-708, which are predicted to regulate the *Nrf2* gene by binding to the 3′-UTR of the mRNA, in regulating the signaling pathway under oxidative stress conditions. Furthermore, we aimed to investigate exogenous modulation of *Nrf2* by supplementation of Que via either directly modulating the *Nrf2* activity or by modulating the expression of the aforementioned candidate miRNAs and its ability to counteract hazardous effects of oxidative stress conditions in bovine granulosa cells.

## 2. Results

### 2.1. Nrf2 is Targeted by miR-153, miR-28 and miR-708

In-silico analysis indicated the conserved binding site of miR-153, miR-28 and miR-708 are located between 105–112 and 58–65 of bovine *Nrf2* 3′-UTR (NM_001011678) (Figure 1A). The luciferase firefly activity was significantly reduced in cells co-transfected with the wild type plasmid constructs and bta-miR-153, miR-28 or miR-708 mimics compared to mutant and negative control plasmids co-transfected with miRNA mimics (Figure 1B).

### 2.2. Oxidative Stress Condition Induced the Expression of Nrf2 and Suppressed the Expression of miR-153, miR-28 and miR-708 in Bovine Granulosa Cells

The effect of oxidative stress on the expression of *Nrf2* and its associated candidate miRNAs was investigated in granulosa cells. Results showed that H_2_O_2_ treatment activated the expression of *Nrf2* (Figure 1C), while the expression of miRNAs targeting *Nrf2* namely; miR-153, miR-28 and miR-708 was reduced significantly (Figure 1D). 

### 2.3. Overexpression of miR-153, miR-28 and miR-708 Suppressed the Expression of Nrf2 and Its Downstream Antioxidants in Bovine Granulosa Cells

Transfection of granulosa cells with the mimics of the candidate miRNAs has led to significant reduction in the expression of Nrf2 and its downstream antioxidant genes (*NQO1* and *PRDX1*) (Figure 2A). Moreover, the protein abundance of *Nrf2* was significantly decreased in cells transfected with miR-153, miR-28 and miR-708 mimics compared to the negative controls (Figure 2B). However, no alteration in the mRNA and protein expression level of *Nrf2* and its downstream target transcripts was observed in cells treated with miRNA inhibitors (Appendix A). 

### 2.4. Overexpression of miR-153 or miR-28 and miR-708 Increased Intracellular ROS Level, Reduced Mitochondrial Activity and Cell Proliferation Rate in Bovine Granulosa Cells

To determine whether deregulation of *Nrf2* levels due to overexpression of miR-153, miR-28 and miR-708 is accompanied by the corresponding oxidative stress phenotypes, the intracellular ROS, mitochondrial activity and cellular proliferation were determined following overexpression of the candidate miRNAs. Results revealed that reduction of *Nrf2* expression via miR-153, miR-28 and miR-708 resulted in a significant increase in intracellular ROS compared to the negative control, accompanied by lower mitochondrial activity (Figure 3A), and reduced the rate of granulosa cell proliferation (Figure 3B).

### 2.5. Selective Knockdown of Bovine Nrf2 Impaired Bovine Granulosa Cell Functions

In order to cross-validate the regulatory role of miR-153, miR-28 and miR-708 in *Nrf2* suppression, the *Nrf2* expression was selectively suppressed using siRNA. Cells co-transfected with siRNA-Nrf2 showed a significant reduction of Nrf2 mRNA and protein level compared to negative control (Figure 4A,C). 

The reduction of *Nrf2* expression level resulted in reduced the expression of downstream antioxidant genes (Figure 4A), and cell proliferation rate (Figure 4B), increased intracellular ROS level and reduced mitochondrial activity (Figure 4D).

### 2.6. Overexpression of miR-153, miR-28 and miR-708 Under Oxidative Stress Negatively Impact on Bovine Granulosa Cell Functions

The consequence of modulation of *Nrf2* targeting miRNAs for cells defense mechanisms was investigated under oxidative stress condition. As expected, overexpression of miRNAs targeting the *Nrf2* gene decreased both the mRNA and protein levels of *Nrf2* and its downstream antioxidants (Figure 5A,B). Moreover, overexpression of the candidate miRNAs resulted in induction of ROS and reduction of mitochondrial activity (Figure 6A) and reduced cellular proliferation rate (Figure 6B).

### 2.7. Quercetin Enhanced Bovine Granulosa Cell Functions under Oxidative Stress Conditions by Inducing the Nrf2 Expression and its Downstream Antioxidants

Treatment of cultured granulosa cells with Que resulted in a significant increment in the expression of *Nrf2* and its downstream antioxidant genes in dose dependent manner (Appendix A). Moreover, supplementation Que at a dose of 10 µM showed significant increase on the expression of Nrf2 protein (Appendix A), slight reduction on intracellular ROS level (Appendix A), increasing mitochondrial activity (Appendix A), slightly enhanced the cellular proliferation rate and the cell cycle (Appendix A). Thus, Que at a dose of 10 µM was selected for further analysis under oxidative stress condition. 

To assess the rescuing role of Que, bovine granulosa cells exposed to oxidative stress for 40 min were then treated with 10 µM Que. Results showed that treatment of cells with Que resulted in upregulation of *Nrf2* (Figure 7A,B), accompanied by increased cellular proliferation rate (Figure 8A), reduced intracellular ROS level (Figure 8B) and increased mitochondrial activity (Figure 8C) in H_2_O_2_-Que treated cells compared to H_2_O_2_ treated alone, suggesting the rescuing effect of Que against oxidative stress condition through activation of *Nrf2* signaling pathway.

Furthermore, the induced expression of *Nrf2* gene by Que in cultured granulosa cells was accompanied by concomitant reduction in the expression of miR-153, miR-28 and miR-708 (Figure 9). 

## 3. Discussion

Mammalian ovary is a metabolically active organ, which generates excess amount of ROS during the final stages of follicular development and ovulation [71,72]. ROS have both positive and deleterious effect in mammalian ovaries. Physiological level of ROS act as a signaling transducer required for growth factor signaling transduction and physiologic adaptation phenomena [73,74,75], oocyte maturation [76,77], ovarian steroid biosynthesis [78] and intermediate decisive changes in cumulus cells prior to ovulation [72]. Moreover, oxidative stress conditions showed mitochondrial morphologic and degenerative changes in mouse granulosa cells. In addition, oxidative stress significantly reduced plasma progesterone and testosterone level, leading to fertility problems in mice [79]. In our previous study, we showed the crucial role of Nrf2 and its downstream antioxidants for the survival of bovine granulosa cells [28] and preimplantation embryos [27] cultured in vitro under oxidative stress conditions. Others have evidenced the correlation between suppression of Nrf2 and inhibition of cellular proliferation [80,81], pronounced increase in ROS production [82,83] and lower mitochondrial activity [83]. Here we have evidenced the presence of several endogenous and exogenous regulatory mechanisms in cultured bovine granulosa cells exposed to oxidative stress.

Target gene prediction tools indicate that more than 85 miRNAs including, miR-144, miR-93, miR-29b-1, miR-153, miR-365-1/miR-193b cluster, miR-28, miR-27-a and miR-142-5p potentially target the *Nrf2* gene [29,84,85]. Previous studies reported the involvement of miR-28 and miR-153 in the Nrf2 regulatory network in different cell lines [50,51,52,53,54]. Moreover, miRNAs are known to be differentially expressed in cell- and tissue-specific manners in various species [32,33,34,35]. Taking these findings into consideration and based on in silico analysis, miR-153, miR-28 and miR-708 were investigated for their role in regulating *Nrf2* under oxidative stress conditions in bovine granulosa cells. Treatment of bovine granulosa cells with H_2_O_2_ increased the expression of *Nrf2* gene accompanied by reduction in expression of miR-153 and miR-28 and miR-708 (Figure 1). This is in agreement with previous studies showing an inverse relationship between *Nrf2* expression pattern and miR-28 in breast carcinoma cell lines [50], miR-93 within rat models of breast carcinogenesis [86] and miR-153 in breast cancer cell lines [51], neuroblastoma cells [52,53] and glioma stem cells [54]. Modulation of miR-153, miR-28 and miR-708 expression individually resulted in reduction of the expression of *Nrf2* both at mRNA and protein level with concomitant reduction in expression its downstream antioxidant genes. The negative interaction between the candidate miRNAs and the *Nrf2* gene was validated by the selective knockdown of the *Nrf2* gene using siRNA designed to target the *Nrf2* gene. Similar phenotypes could be observed between overexpression of the candidate miRNAs and siRNA targeting the *Nrf2* gene with respect to the expression of the *Nrf2* gene both at mRNA and protein level, ROS accumulation, cellular proliferation, mitochondrial activity and the expression of antioxidants. Over accumulation of intracellular ROS level was associated with reduction in mitochondrial activity [28]. Our results were in agreement with the aforementioned results, where overexpression of miR-153, miR-28 and miR-708 led to an increment in the level of intracellular ROS and impaired mitochondrial activity accompanied by a reduction in cellular proliferation. Overexpression of candidate miRNAs under oxidative stress condition showed a reduction in *Nrf2* mRNA and protein levels resulted in increasing intracellular ROS level and reduced mitochondrial activity which led to lower cellular proliferation compared to H_2_O_2_ alone treated group. This has evidenced the potential of modulating the *Nrf2* mediated oxidative stress response mechanisms in bovine granulosa cells by inducing the expression of the candidate miRNAs. The potential of this approach in therapeutic applications could be the focus of future research.

Oxidative stress affects cattle reproduction and several metabolic processes in transition cows. Supplementation of antioxidants is reported to induce protective effects against oxidative stress conditions and restores cellular antioxidant defense mechanisms [87], which showed beneficial effect on the quality of bovine meat and milk production [88]. Quercetin is a member of polyphenolic compounds known as flavonoids. Accumulating evidence demonstrated that Que have an antioxidant activity against oxidative stress in different cell types [89,90]. Furthermore, dietary supplementation of Que in heat stressed rabbit showed improved follicular development, reduced apoptosis in granulosa cells and maintained oocyte competence [91]. Several studies showed that Que exert an anti-inflammatory effect through inhibition of *NF-κB* and activation of *Nrf2* [64,65,66]. In the present study, supplementation of Que to cultured granulosa cells resulted in increased transcription of *Nrf2* in dose dependent manner. Low dose of Que has resulted in increasing *Nrf2* at mRNA and protein levels accompanied by increased cellular proliferation and mitochondrial activity. However, higher concentrations of Que resulted in increased intracellular ROS level, reduce the mitochondrial activity and induced S-phase cell cycle arrest. These findings are in agreement with previous reports on the impact of Que on *Nrf2* in different cellular functions in various cell models [92,93,94]. On the other hand, Que has the capability to act as pro or antioxidant depending on concentration and cellular model [95]. The antioxidant activity of Que was reported at low concentration, while higher concentrations resulted in decrease cellular proliferation as a result of induced apoptosis [96], stimulate the generation of superoxide radicals (O_2_^•−^) and subsequently affecting mitochondrial activity [97]. Moreover, Que is reported inducing a cell cycle arrest at G0/G1, S-phase or G2/M depending on cell types [98,99,100,101]. Similarly, our findings showed that Que treatment at lower concentration resulted in cellular viability enhancement, reduction of intracellular ROS level and increased mitochondrial activity, as it has been reported before [62,102,103]. Que is known to modulate expression of *Nrf2* through multiple pathways including transcriptional regulation, posttranscriptional through stabilizing Nrf2 protein and inhibiting *Nrf2* ubiquitination and post-translational level through Keap1 modification [104]. Similarly, our results revealed for the first time that Que could modulate the expression of miR-153, miR-28/708 expression pattern. This could show the indirect effect of Que on *Nrf2* activity in addition to its direct effect. Such mechanisms of action of flavonoids including Que has been reported in vivo and in vitro experimental setups [105,106,107,108,109]. Taking all of the results together, a hypothetical mechanism of both endogenous and exogenous modulation of the *Nrf2* mediated oxidative stress defense mechanism in bovine granulosa cells is illustrated in Figure 10.

## 4. Materials and Methods

### 4.1. Bovine Granulosa Cell Culture

Bovine ovaries were collected and transported from local slaughterhouse in thermo-flask containing warm (37 °C) physiological saline (NaCl 0.9%) solution. Upon arrival, ovarian sample were washed with calcium and magnesium free phosphate buffer saline (PBS-CMF) three times followed by rinsing in 70% ethanol for 30 s. Thereafter, the follicular fluid was aspirated from small growing follicles (3–5 mm diameter) using a 20-gauge needle and collected on pre-warmed PBS-CMF in 15 mL tube. After collection, the aspirated follicular fluid was kept in standing position for 15 min at 37 °C to allow the cumulus-oocyte-complex (COC) to settle at the bottom of the tube. The upper supernatant part containing the granulosa cells were transferred to another 15 mL tube and centrifuged at 750 rpm for 7 min. The pellets were resuspended in red blood cells (RBCs) lysis buffer for 1 min, followed by addition of DMEM/F12-HAM (Sigma-Aldrich, München, Germany) to stop lysis buffer reaction and centrifuged at 500 rpm for 7 min. Trypan blue exclusion method was used to determine cell viability and concentration. Cells were cultured at rate of 2.5 × 10^5^/well in CytoOne^®^-24 well plate (Starlab International GmbH, Hamburg, Germany) in 500 μL DMEM/F12-HAM supplemented with 10% fetal bovine serum (FBS) (Gibco FBS, Life technologies, Schwerte, Germany), 100 IU/mL penicillin, 100 µg/mL streptomycin and 100 µg/mL fungizone (Sigma-Aldrich, München, Germany) and incubated at 37 °C in and 5% CO_2_.

### 4.2. MicroRNA Target Gene Prediction and Luciferase Reporter Assay

Prediction of miRNAs targeting Nrf2 was done using TargetScan; an online target prediction database (http://www.targetscan.org), miRNAs (miR-153, miR-28 and miR-708) were selected based on the probability of preferential conservation [110,111].

The interactions between miR-153, miR-28 and miR-708 and *Nrf2* gene were validated using luciferase reporter assay as described in Gebremedhn et al. [112]. Briefly, DNA fragments containing the putative miRNA binding sites in the 3′-UTR of *Nrf2* gene (wild type) for the aforementioned miRNAs were amplified from bovine genomic DNA. Similarly, DNA fragment containing mutations at the binding sites (mutant) were separately designed. The designed wild type and mutant fragments were cloned into pmirGLO Dual-Luciferase miRNA Target Expression Vector (Promega GmbH, Mannheim, Germany) between the *SacI* and *Xhol* restriction sites (Appendix A). The presence of the miRNA binding sites in the wildtype construct and the absence of the binding sites in the mutant constructs were confirmed by sequencing the PCR amplicon of the pmirGLO vector.

Following this, sub-confluent in vitro cultured granulosa cells (70–80% of confluency) were co-transfected with 350 ng of plasmid containing either wild-type or mutant sequence with 100 nM of the corresponding miRNA mimics (Exiqon, Vedbaek, Denmark) using Lipofectamine^®^ 2000 transfection reagent (Life Technologies, Germany). Twenty-four hours later, cells were lysed using 1x Passive Lysis Buffer (Promega GmbH, Germany) and the firefly and Renilla Luciferase activities were determined based on the Dual-Luciferase^®^ Reporter (DLR™) Assay System (Promega GmbH, Germany) according to manufacturer’s protocol. The absorbance of firefly and Renilla luciferase activity were measured using Centro LB 960 Microplate Luminometer (Berthold Technologies GmbH, Bad Wildbad, Germany). Data was analyzed as the ratio of firefly to Renilla activity.

### 4.3. MicroRNA and siRNA Transfection

To determine the impact of candidate miRNAs in modulating the expression of Nrf2, candidate miRNAs were either overexpressed or inhibited using chemically synthetized miRNA mimics and inhibitors (Exiqon, Denmark), respectively. For this, 100 nM miRNA mimic, inhibitor, or the corresponding negative control (NC) were transfected in sub-confluent granulosa cells using Lipofectamin^®^ 2000 (Invitrogen, Carlsbad, CA, USA) transfection reagent in Opti-MEM I reduced-serum medium (Invitrogen, Carlsbad, CA, USA). Twenty-four hours post-transfection, cells were subjected to mRNA and protein expression analysis. Moreover, detection of intracellular ROS level, cell proliferation assay and assessment of mitochondrial activity were performed.

In order to cross-validate the regulatory role of candidate miRNAs on *Nrf2* gene functions, targeted knockdown of Nrf2 was performed using bovine specific siRNA (Exiqon, Denmark). For this, sub-confluent granulosa cells were transfected with 200 nM of siRNA -Nrf2 or siRNA negative control (NC) using Lipofectamin 2000 in Opti-MEM I reduced-serum medium. Twenty-four hours post-transfection, the cells were subjected to intracellular ROS level detection, cell proliferation assay, assessment of mitochondrial activity, mRNA and protein expression pattern analyses.

### 4.4. Exogenous Induction of Nrf2 by Quercetin

Stock solutions of Que were freshly prepared by dissolving it in Dimethyl sulfoxide (DMSO). The stock solutions were subsequently diluted with Dulbecco’s Modified Eagle Medium: Nutrient Mixture F-12 (DMEM/F12-HAM) media. The final concentrations of DMSO in the medium were ≤0.01% (*v*/*v*). Appropriate controls with only the vehicle DMSO were included in all experiments. For exogenous Nrf2 modulation, the appropriate and non-toxic dose of Que was determined. For that, sub-confluent granulosa cells were treated with different concentrations of Que (0, 10, 20, 50, 100 or 200 µM) for 24 h. Treated cells were then subjected to intracellular ROS level detection, cell cycle assay, cell proliferation assay and assessment of mitochondrial activity. Moreover, mRNA and protein expression patterns of candidate genes were quantified in each treatment group.

### 4.5. Induction of Oxidative Stress Using H_2_O_2_

The effects of miR-153, miR-28 and miR-708 overexpression and the rescuing role of Que on bovine granulosa cells when subjected to oxidative stress condition were investigated. For this, cells were treated with 5 µM H_2_O_2_ for 40 min [28], then transfected with miRNA mimics or supplemented with 10 µM Que. Twenty-four hours post-treatment, cells were subjected to intracellular ROS level, mitochondrial activity, cellular proliferation rate and Nrf2 expression pattern analyses.

### 4.6. Total RNA Isolation and Quantitative Real-Time PCR (qRT-PCR)

Cells were harvested 24-h post-treatment (miRNA, siRNA transfection and que supplementation), subjected to total RNA isolation using miRNeasy^®^ mini kit (Qiagen GmbH, Hilden, Germany) following manufacturer’s protocol. After assessing the quality and concentration of the RNA samples using NanoDrop 8000 spectrophotometer (NanoDrop technologies, Schwerte, Germany), cDNA synthesis was performed using first stand cDNA synthesis kit (Thermo Fisher scientific, Schwerte, Germany). Briefly, RNA concentration was adjusted using nuclease-free water to total volume 10 µL from each replicate was followed by co-incubation with 0.5 µL oligo (dT)18 and 0.5 µL random primer at 65 °C for 5 min. Next, 1 µL RiboLock, 4 µL 5× reaction buffer, 2 µL dNTPs and 2 µL reverse transcriptase were added for each sample and co-incubated at 25 °C for 5 min, 37 °C for 60 min, and 70 °C for 5 min. After incubation, samples were stored at −20 °C till gene expression analysis.

The relative abundance of Nrf2 and its downstream antioxidant genes; *NQO1, PRDX1, SOD1* and *CAT* was quantified using iTaq™ Universal SYBR^®^ Green Supermix (Bio-Rad Laboratories GmbH, München, Germany) in Applied Biosystem^®^ StepOnePlus™ (Applied Biosystems, CA, USA) using gene specific primers (Appendix A). All primers were designed using NCBI primer designing tool (http://www.ncbi.nlm.nih.gov/tools/primer-blast/). Data was analyzed using the comparative Ct (2^−ΔΔCt^) methods (Livak and Schmittgen 2001) and the average expression levels of *ACTB* and *GAPDH* was used for normalization.

For miRNA expression analysis, cDNA was synthetized using miRCURY^®^ LNA^®^ RT kit (Qiagen GmbH, Germany) following the manufacturer’s protocol. Briefly, 80 ng of miRNA-enriched total RNA was used for cDNA synthesis. Reverse transcription master mix of 2 µL reaction buffer, 1 µL enzyme mix, and complete reaction volume up to 10 µL. Thereafter, incubate samples at 42 °C for 60 min and 95 °C for 5 min to inactivate the reverse transcription enzyme. Synthetized cDNA was 15× diluted and used for RT-PCR analysis using miRCURY LNA SYBR^®^ Green PCR kit (Qiagen GmbH, Germany) following the manufacturer’s protocol. The thermal cycler was programmed for initial preheating at 95 °C for 2 min, followed by 40 cycles of amplification at 95 °C for 10 s and 56 °C for 60 s followed by melting curve analysis. Data was analyzed using comparative Ct (2^−ΔΔCt^) methods [113] and the expression level of 5S ribosomal RNA and U6 were used as internal control for normalization.

### 4.7. Cell Proliferation Assay

To determine the impact of modulation of Nrf2 by the candidate miRNAs on cell proliferation, 1.5 × 10^4^ granulosa cells were seeded into 96-well plate and cultured in the F-12 media. Sub-confluent granulosa cells were then transfected with miRNA, siRNA or treated with different concentration of Que. Twenty-four hours post-treatment, 10 μL of CCK-8 kit solution (Dojindo EU GmbH, München, Germany) was added into each well and the plate was incubated for 4 h at 37 °C in and 5% CO_2_. The optical density (OD) was at a wavelength 450 nm using Synergy™ H1 Multi-Mode Reader (BioTek, Bad Friedrichshall, Germany). OD from empty wells was used for background correction.

### 4.8. Protein Immunofluorescence Detection

The abundance and localization of the Nrf2 protein in granulosa cells subjected to transfection with miRNA, siRNA and Que was determined using immunocytochemistry. Briefly, cells cultured in 8-well chamber slide. 24 h post treatment, cells were washed with PBS-CMF, and then fixed overnight at 4 °C in 4% (*w*/*v*) paraformaldehyde in PBS-CMF. Fixed cells were washed three times with PBS-CMF, then permeabilized with 0.3% (*v*/*v*) Triton-X100 (Sigma-Aldrich) for 10 min at room temperature followed by washing with PBS-CMF 3 times for 5 min. Cells were incubated in 4% donkey serum (Sigma-Aldrich) for 1 h at room temperature, followed by incubation overnight at 4 °C with polyclonal rabbit primary antibodies against Nrf2 (1:100, orb11165, Biorbyt, Cambridge, UK). Then, cells were further incubated at 37 °C for 3 hr in the dark with fluorescence-labelled secondary antibody (Alexa flour™ 568 goat anti-rabbit 1:350, Life Technologies, Germany). A droplet of Vectashield mounting medium containing DAPI (Dabco; Acros, Geel, Belgium) was used to stain the nuclei. Finally, images were visualized under a CLSM LSM-780 confocal laser-scanning microscope (Carl Zeiss GmbH; Jena, Germany) and analyzed using ImageJ 1.48v (National institutes of Health, Maryland, USA, https://imagej.nih.gov/).

### 4.9. Assessment of Mitochondrial Activity

The mitochondrial activity in bovine granulosa cells transfected with miRNA, siRNA and Que was assessed using MitoTracker^®^ Red CMXRos (M7512; Invitrogen) according to manufacturer′s instructions. Breifly, lyophilized MitoTracker^®^ product was dissolved in a high-quality, anhydrous dimethylsulfoxide (DMSO) to a final concentration of 1 mM. The final working concentration (200 nM) was prepared by diluting stock solution in DMEM/F12-HAM medium. MitoTracker^®^ Red CMXRos is a red-fluorescent dye that stains mitochondria in live cells and its accumulation is dependent upon membrane potential (MMP), which is a marker for mitochondrial functionality [114,115]. For that, granulosa cells were cultured in eight-well slide and cells were incubated with 200 nM MitoTracker^®^ red dye at 37 °C for 30 min, followed by two washing with PBS-CMF and then fixed overnight at 4 °C with 4% paraformaldehyde. Fixed cells were mounted with Vectashield (H-1200) containing DAPI. Images were acquired at 40x magnification under a CLSM LSM-780 confocal laser-scanning microscope (Carl Zeiss GmbH; Germany) and analyzed using ImageJ 1.48v (National institutes of Health, Maryland, USA, https://imagej.nih.gov/).

### 4.10. Cell Cycle Assay

The cell cycle status of cells treated with different doses of Que was determined using propidium iodide (PI) staining in flow cytometer as previously described in [37,38,39]. Briefly, cells were trypsinized 24 h after treatment. Following cell counting, a minimum of 1 × 10^6^ cells were fixed overnight at 4 °C in ice-cold 70% ethanol. Fixed cells were stained with 50 µg/mL PI (Invitrogen, Carlsbad, CA, USA) and 50 μg/mL RNase and readings were acquired in flow cytometer (BD Biosciences FACS Calibur, CA, USA). A minimum of 10,000 cells were acquired per sample and data were analyzed using ModFit LT software (http://www.vsh.com/products/mflt/index.asp).

### 4.11. Intracellular ROS Detection

Cultured cells in each experiment were subjected to ROS accumulation assay using 2′,7′-dichlorofluorescin diacetate (H2DCFDA) (Life Technologies, Germany) according to manufacturer′s instructions. Cells were incubated with 50 µL of 75 µM H2DCFDA diluted in PBS-CMF for 20 min in dark at 37 °C. Following this, cells were washed twice in PBS-CMF, and images were captured immediately under an inverted fluorescence microscope (Leica DM IRB, Leica, Wetzlar, Germany) using a green-fluorescence filter and images were analyzed using ImageJ 1.48v (National institutes of Health, Maryland, USA, https://imagej.nih.gov/).

### 4.12. Statistical Analysis

Data were analyzed using GraphPad Prism 5 (GraphPad, San Diego, CA, USA) and presented as mean ± SEM of at least three biological replicates. Statistical significance between mean values of more than two treatment groups was determined using one-way analysis of variance (ANOVA) followed by Tukey multiple pairwise comparison. Moreover, data from two treatment groups were analyzed using student′s two-tailed t-test. The statistical significance was determined at *p* = 0.05.

## 5. Conclusions

In conclusion, the present study evidenced the mechanisms of regulation of Nrf2 mediated oxidative stress response pathway in bovine granulosa cells and indicate the potential application of those regulatory mechanisms in future fertility treatment strategies to enhance ovarian functionality.

## Figures and Tables

**Figure 1 ijms-20-01635-f001:**
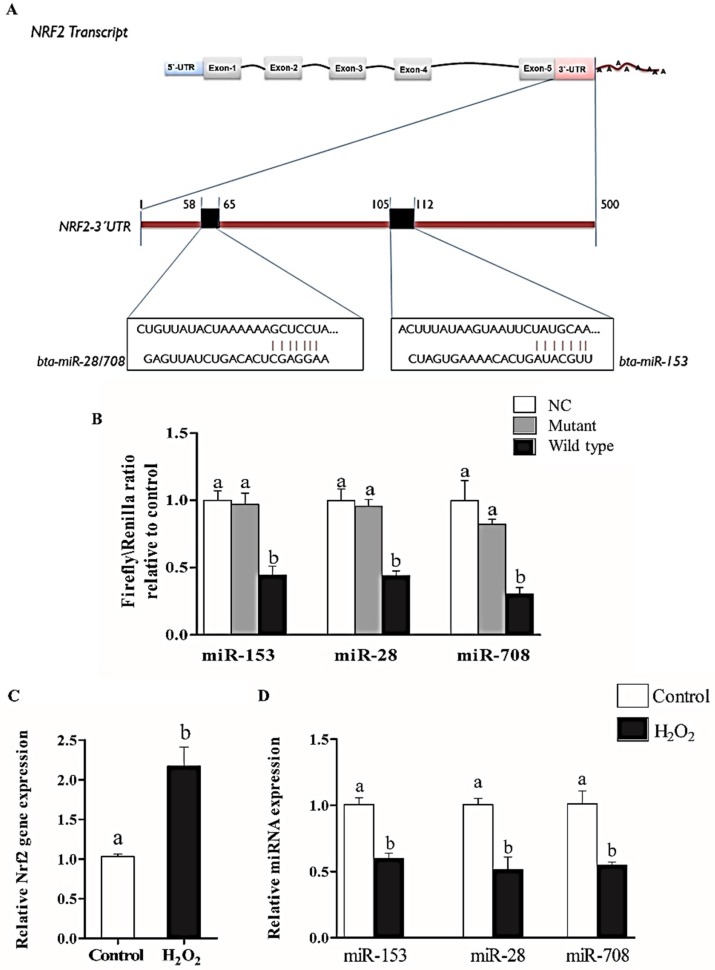
Conserved binding site of miR-153, miR-28 and miR-708 are located between 105–112 and 58–65, respectively in bovine *Nrf2* 3′-UTR (**A**). The luciferase firefly activity is reduced in granulosa cells co-transfected with bta-miR-153, miR-28 and miR-708 mimics with the wild type pmiRGlo expression vector (**B**). H_2_O_2_-induced oxidative stress increased cellular mRNA expression level of *Nrf2* (**C**), as well as decreased miR-153, miR-28 and miR-708 expression level (**D**). qRT-PCR analysis of *Nrf2*, miR-153, miR-28 and miR-708 in granulosa cells under normal condition (white bar) and oxidative stress condition (black bar). Data are presented mean ± SEM of three independent biological replicates. Bars with different letters (a,b) showed statistically significant differences (*p* < 0.05).

**Figure 2 ijms-20-01635-f002:**
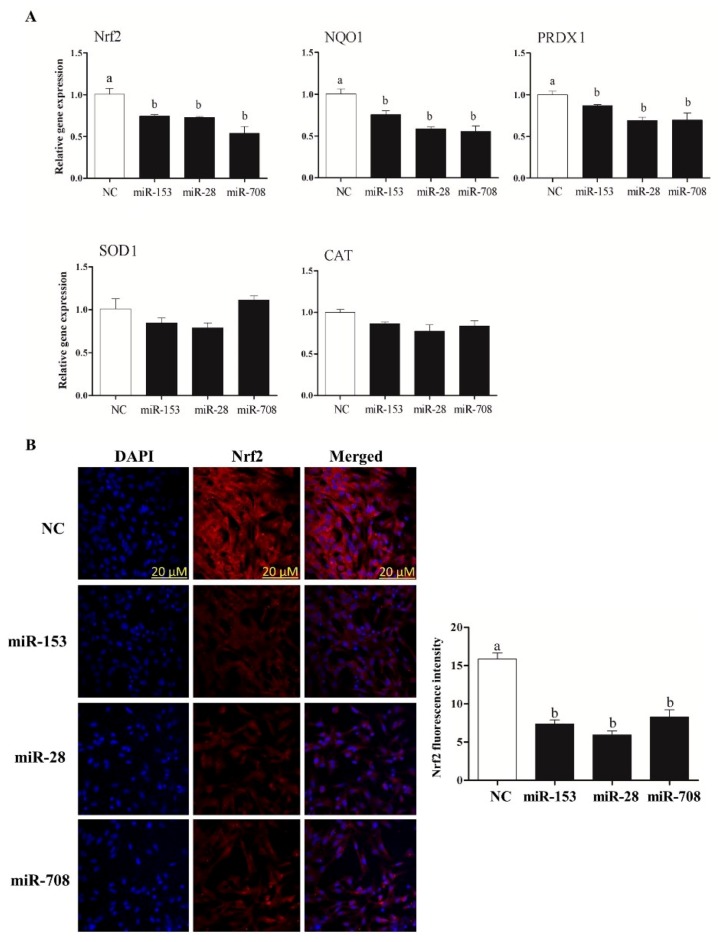
Quantitative RT-PCR of *Nrf2* and its downstream antioxidants in bovine granulosa cells co-transfected with miR-153, miR-28 and miR-708 mimics (**A**). Immunocytochemistry of Nrf2 in bovine granulosa cells co-transfected with miR-153, miR-28 and miR-708 mimics (**B**). Data are presented as mean ± SEM of three independent biological replicates. Bars with different letters (a,b) showed statistically significant differences (*p* < 0.05).

**Figure 3 ijms-20-01635-f003:**
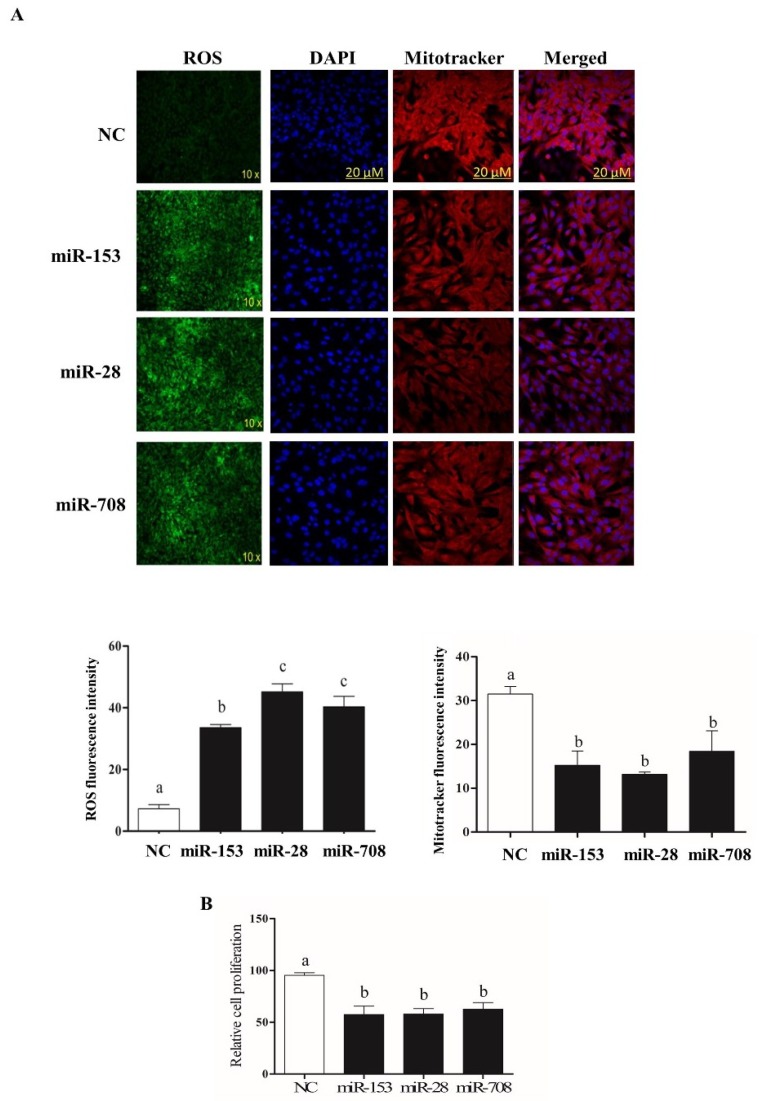
Higher level of intracellular Reactive oxygen species (ROS) level accompanied with lower mitochondrial activity in bovine granulosa cells co-transfected with miR-153, miR-28 and miR-708 (**A**). Moreover, overexpression of miR-153, miR-28 and miR-708 resulted in reduced bovine granulosa cell proliferation (**B**). White bar indicates the proliferation rate of the negative control and black bars represent proliferation rate of cells co-transfected with miRNA mimics. Data are presented as mean ± SEM of three independent biological replicates. Bars with different letters (a,b) showed statistically significant differences (*p* < 0.05).

**Figure 4 ijms-20-01635-f004:**
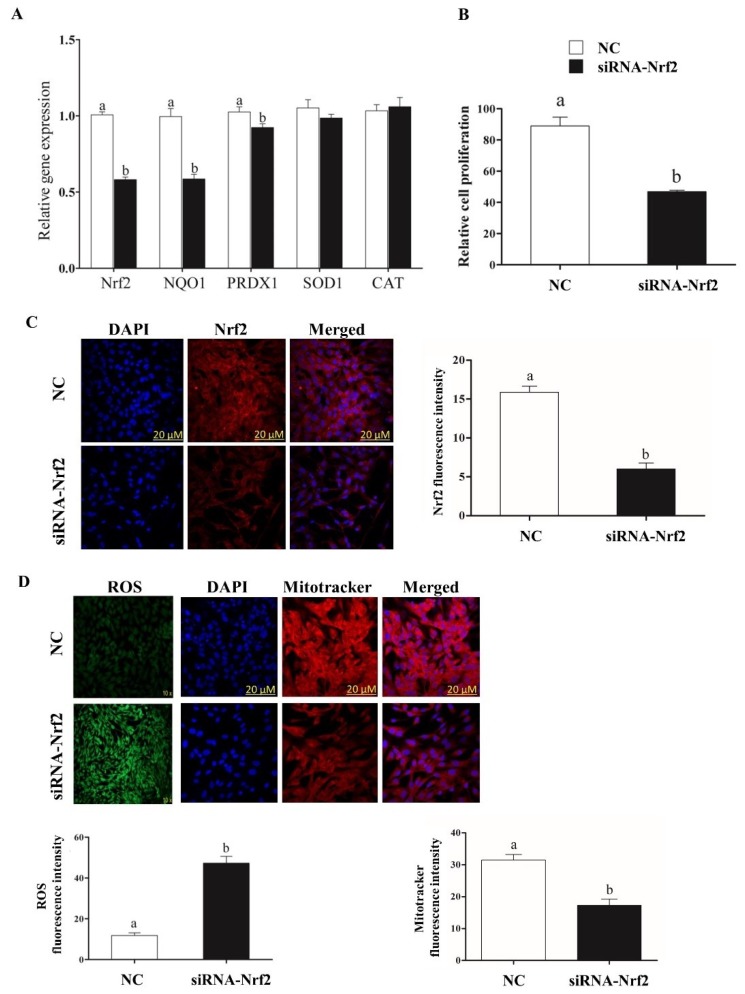
Selective knockdown of *Nrf2* reduced cellular mRNA expression level of *Nrf2* and its downstream antioxidant genes (**A**). Selective knockdown of *Nrf2* reduced bovine granulosa cell proliferation (**B**). Immunocytochemistry of *Nrf2* in bovine granulosa cells co-transfected with siRNA targeting the *Nrf2* (**C**). Moreover, higher intracellular ROS level and reduced mitochondrial activity (**D**) was observed in Nrf2-knockdown cells compared to the control counterparts. White bars indicate the cells transfected with the NC and the dark bars indicate cells transfected with siRNA targeting Nrf2. Data are presented as mean ± SEM of three independent biological replicates. Bars with different letters (a,b) showed statistically significant differences (*p* < 0.05).

**Figure 5 ijms-20-01635-f005:**
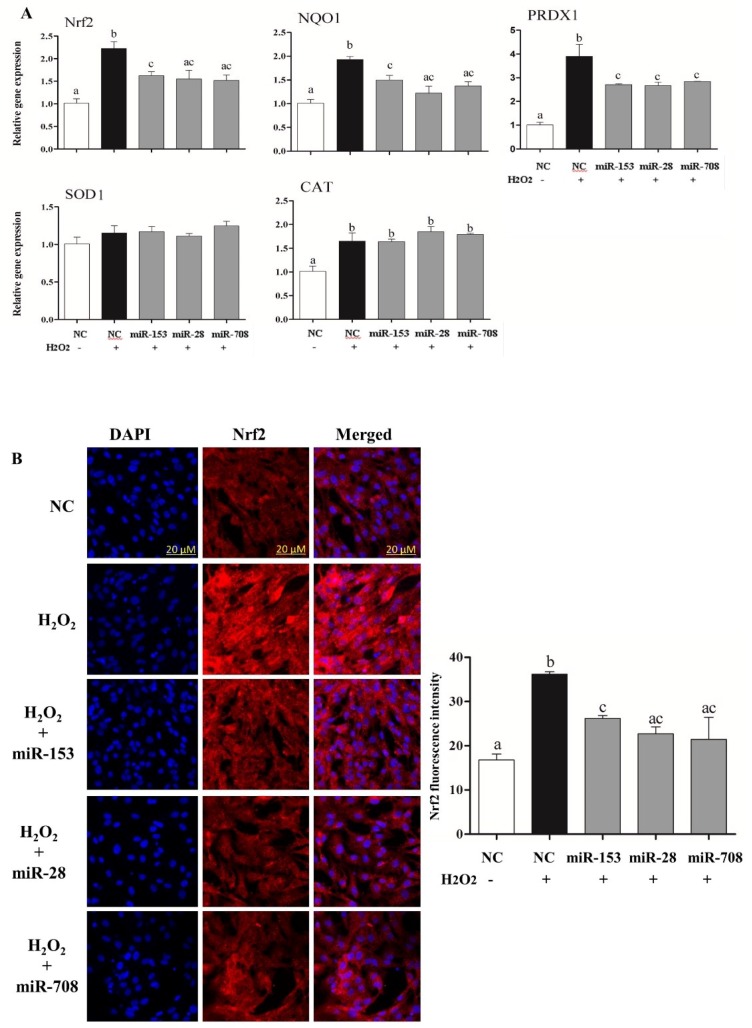
Overexpression of miR-153, miR-28 and miR-708 under oxidative stress conditions reduced mRNA expression level of Nrf2 and its downstream antioxidants(**A**). Immunocytochemistry of *Nrf2* in bovine granulosa cells co-transfected with miR-153, miR-28 and miR-708 mimics under oxidative stress conditions. The Nrf2 protein expression level was reduced under oxidative stress conditions followed by transfection with miR-153, miR-28 and miR-708 mimics individually compared to H_2_O_2_- treated group (**B**). White bars indicate the cells transfected with negative control without H_2_O_2_. Dark bars indicate cells transfected with negative control under H_2_O_2_ and gray bars represent cells transfected with miRNA mimics under the H_2_O_2_ challenge. Data are presented as mean ± SEM of three independent biological replicates. Bars with different letters (a,b,c) showed statistically significant differences (*p* < 0.05).

**Figure 6 ijms-20-01635-f006:**
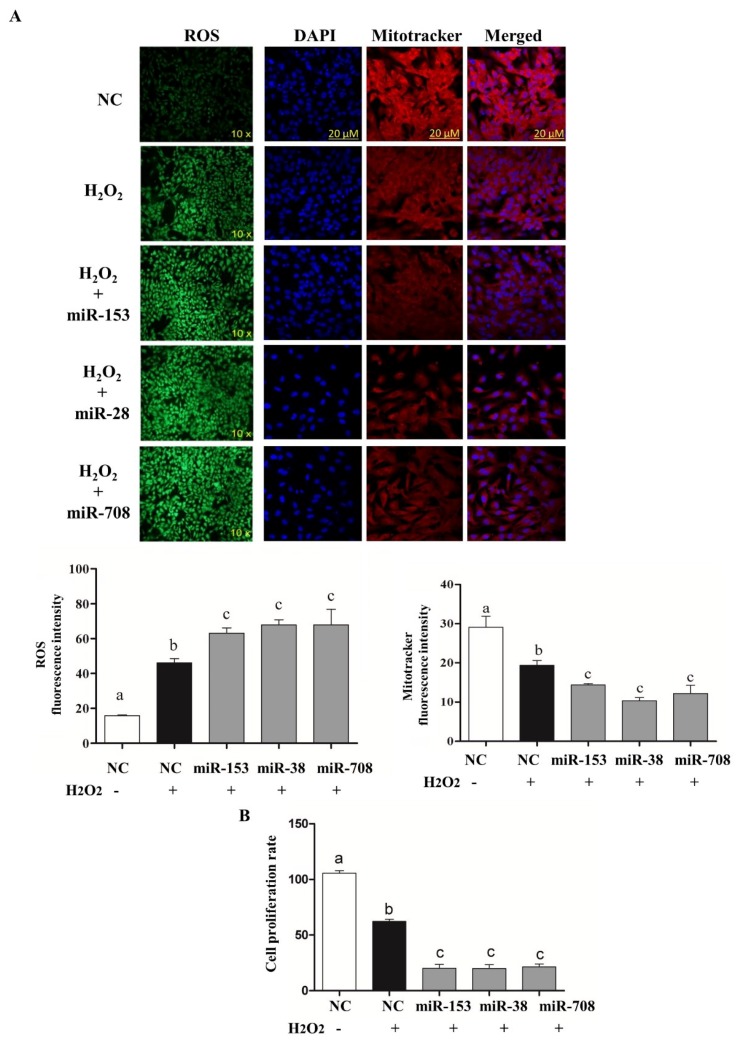
Overexpression of miR-153, miR-28 and miR-708 under oxidative stress conditions increased the intracellular ROS level and reduced the mitochondrial activity compared to cells treated with only H_2_O_2_ (**A**). Moreover, bovine granulosa cell proliferation was significantly reduced in cells transfected with miRNA mimics under H_2_O_2_ (**B**). White bars indicate the cells transfected with negative control without H_2_O_2_. Dark bars indicate cells transfected with negative control under H_2_O_2_ and gray bars represent cells transfected with miRNA mimics under the H_2_O_2_ challenge. Data are presented as mean ± SEM of three independent biological replicates. Bars with different letters (a,b,c) showed statistically significant differences (*p* < 0.05).

**Figure 7 ijms-20-01635-f007:**
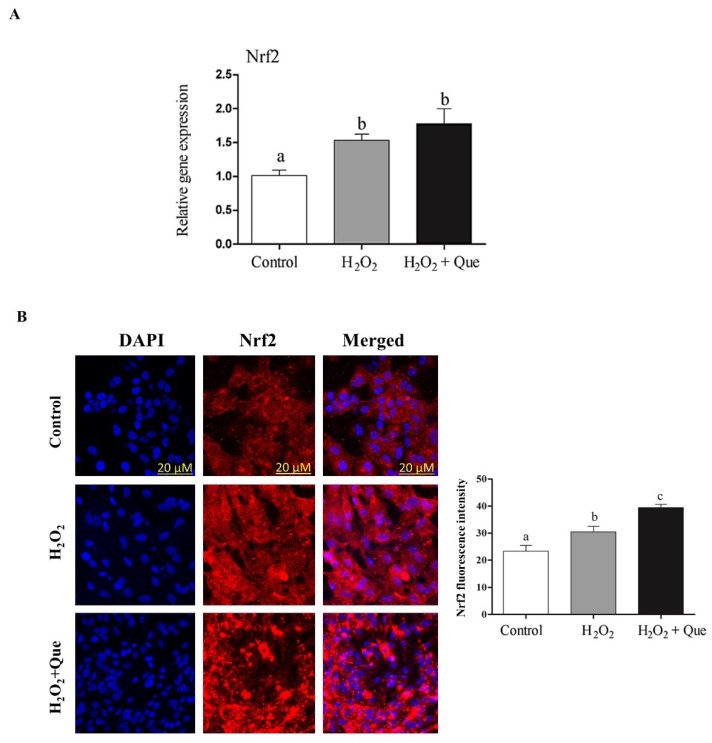
Treatment of cells with quercetin showed slight increment in the expression of *Nrf2* under oxidative stress (**A**). Bovine granulosa cells co-incubated with quercetin under oxidative stress condition showed upregulation of *Nrf2* at protein level (**B**). The white bar indicates the control group and the gray and black bars indicateH_2_O_2_ treated and cells treated with both H_2_O_2_ and quercetin, respectively. Data are presented as mean ±SEM of three independent biological replicates. Bars with different letters (a,b,c) showed statistically significant differences (*p* < 0.05).

**Figure 8 ijms-20-01635-f008:**
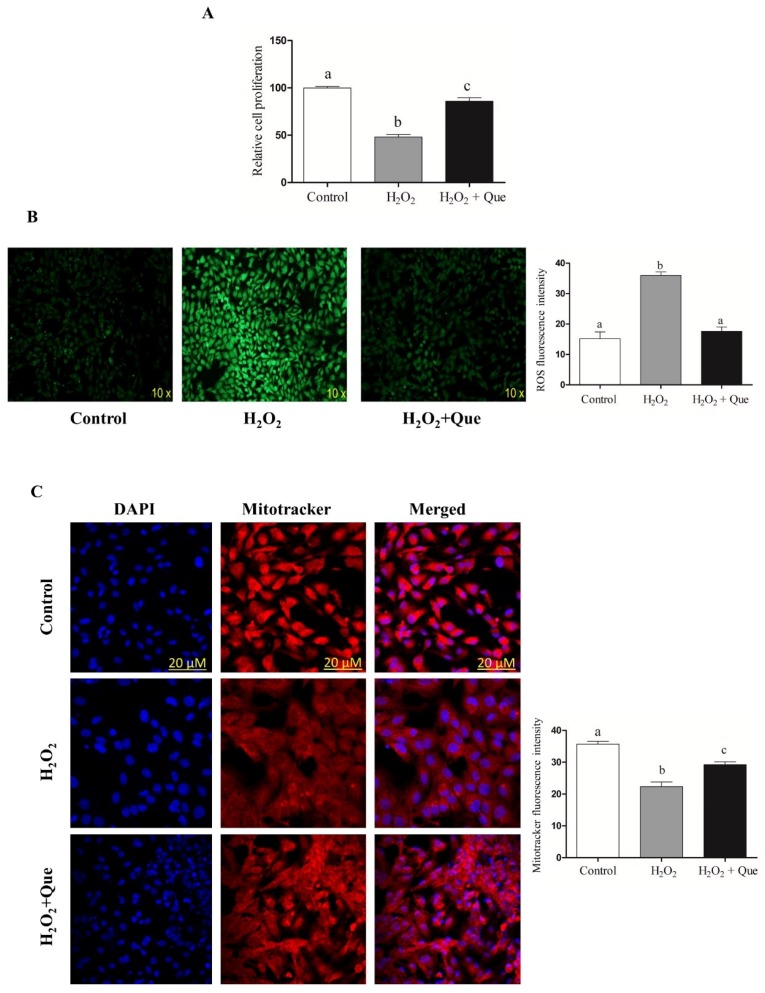
Quercetin rescue bovine granulosa cells from the oxidative stress damage induced by H_2_O_2_. Supplementation of quercetin following H_2_O_2_ treatment increased granulosa cells proliferation (**A**), reduced intracellular ROS level (**B**) and increase mitochondrial activity (**C**) compared to cells treated with only H_2_O_2_. The white bar indicates the control group, while the gray and dark bars represent cells treated with only H_2_O_2_ and cells treated with both H_2_O_2_ and quercetin, respectively. Data are presented as mean ± SEM of three independent biological replicates. Bars with different letters (a,b,c) showed statistically significant differences (*p* < 0.05).

**Figure 9 ijms-20-01635-f009:**
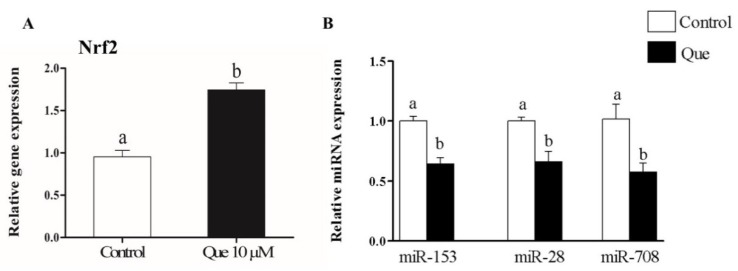
Quercetin increased cellular mRNA expression level of *Nrf2* (**A**). The expression of miR-153, miR-28 and miR-708 was downregulated following quercetin treatment of cells (**B**). White bar represent the control groups and dark bars represent quercetin treated group. Data are presented as mean ± SEM of three independent biological replicates. Bars with different letters (a,b) showed statistically significant differences (*p* < 0.05).

**Figure 10 ijms-20-01635-f010:**
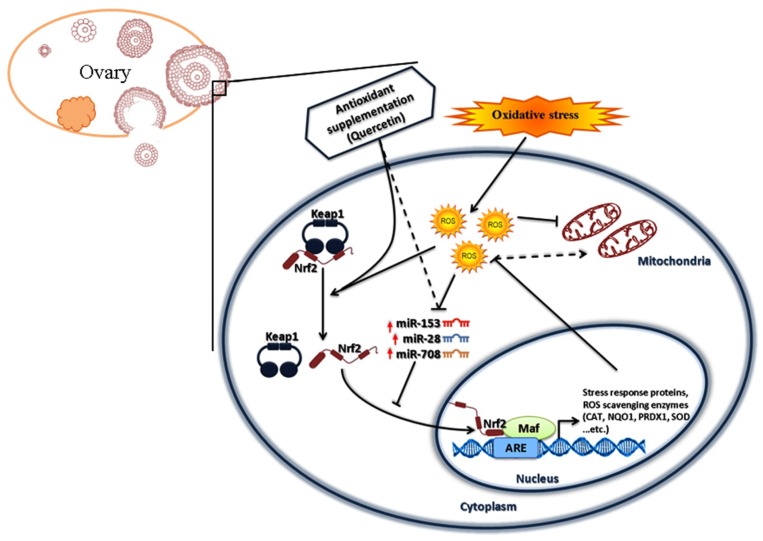
Model of the Nrf2-mediated oxidative stress response pathway in bovine granulosa cells under moderate oxidative stress conditions and under antioxidant (quercetin) supplementation. Oxidative stress and Que modulate *Nrf2* post transcriptionally through the inhibition of miR-153, miR-28 and miR-708, which targeted bovine *Nrf2*. After that, Nrf2 protein was activated and localized to the nucleus, where it binds to the antioxidants reactive element (ARE) in the promoters of antioxidant genes. This activates antioxidant gene expression so the respective proteins can scavenge the excessive ROS and subsequently maintain the activity of mitochondria.

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
