# Peer review of "Endogenous and Exogenous Modulation of Nrf2 Mediated Oxidative Stress Response in Bovine Granulosa Cells: Potential Implication for Ovarian Function"

_ijms, 2019, doi:10.3390/ijms20071635_

Round 1
Reviewer 1 Report
The manuscript by Khdrawy et al reports an ability of three miRNAs (miR-82, miR-153, and miR-708 to suppress expression of the Nrf2 transcription factor and of the Nrf2-responsive genes, and the effect of H2O2 and quercetin on this pathway. The paper is quite clearly written, with just several question to the grammatic issues. The Experimental section is also presented inn a clear way, so that anyone can reproduce the data.
However, there are several questions to the manuscript and the study itself that put into question some of the conclusions. In addition, the rationale of the project is nor clear.
Major questions
1. One of the main conclusions of the study is that Nrf2 is regulated by three miRNAs (82, 153, and 708). However, for two of them it is a well-known fact, since several papers had been already published (and several of them are cited here). So, it should be discussed in the Introduction. And in the introduction the authors need to state more clearly what did they want to study here. It is important to understand the novelty of the study.
2. Usage of H2O2 as a prooxidant: why did you choose to use 10 µM concentration (so low)?
3. Fig 1C and Fig. 9 : quantification of Nrf2 levels should be also done at the protein level (for example, by Western blot), as Nrf2 is subjected to proteolysis by Keap1 and is stabilized upon dissociation. In addition, it should be explained why Nrf2 levels was accessed at mRNA level.
4. Mitotracker assay that was used to access mitochondrial activity. if I am not mistaken, this dye stains thiol groups in mitochondria. Thus, its staining correlates with mitochondria mass and not with its activity. So, it should be checked and revised if needed throughout the text (as it puts into question one of the conclusions). Moreover, I am not sure that the conclusion on Fig. 6A is correct, as H2O2 triggers oxidation of thiol groups. Thus, a decrease in mitotracker staining here may show not a decrease in mitochondria mass but merely oxidation of thiol groups. So, a conclusion needs to be verified by other method as well.
5. Line 148: The conclusion about inadequate mitochondrial distribution is not supported by the figure, as the image was captured at low resolution, and the mitochondrial structure (form, network) is not seen at all. So, images at better resolution are required.
6. Measurement of ROS was performed with H2DCFDA. However, this dye is also oxidized by peroxynitrite, and sometimes its fluorescence depends on redistribution of iron from mitochondria. So, to meet currently required experimental level, these experiments need to be verified with other dyes (such as dihydroethidium or any other).
7. Fig. 5B and 7B: H2O2 triggered elevation of Nrf2 level in the cells, However, basing on the figure it is impossible to say that the transcription factor changed its localization (translocated into the nucleus, as it should have done). And it is a bit strange, taking into account that in these cells there is a clear up-regulation of Nqo1 and HO-1. On Fig 7B it is quite clear that the majority of Nrf2 is outside the nuclei. So, the authors probably should consider analyzing Nrf2 localization (for example, by separating fractions of nuclear and cytoplasmic proteins using NE-PER kit with subsequent Nrf2 measurement by western blotting. It should strengthen the conclusion.
8. Fig. 10. There is an indication that Nrf2-responsive genes stimulate mitochondria (activity, mass??). But strictly speaking, the authors did not access any mechanisms (transcription factor for example) that are known to affect mitochondria biogenesis. As one, Nrf1 (nuclear respiratory factor 1) could be named. So, it could be worth mentioning this limitation in the discussion and pointing out to necessity for such studies in future.
Minor issues
1. Please revise the text. For example, pay attention to the sentence on lines 58-64: it should be in present tense and in third form.
2. Superoxide anion should be written correctly (O2●-), as it is a radical actually. So, revise it in the lines 43 and 249.
3. Line 93: HMOX1 is a gene. If the authors wanted to name the corresponding protein, it should have been HO-1.
4. Fig 2A, 4A, : please state that SOD was SOD 1, and Prdx was Prdx 1. By the way, why of several Prdx isoforms that are regulated by Nrf2, you chose Prdx1.
5. Discussion: In the introduction the authors discussed that Nrf2 controls expression of Nqo1, HO-1, Prdx1, CAT, and SOD1. But the experimental data here show absence of such control for the latter three genes. So it could be mentioned in the discussion.
6. Line 416: Please write in which buffer/media the cells were incubated with H2DCFDA, and what was the final concentration of the dye.
Author Response
We sincerely appreciate the reviewers for the valuable comments
Reviewer #1:
Comment: One of the main conclusions of the study is that Nrf2 is regulated by three miRNAs (28, 153, and 708). However, for two of them it is a well-known fact, since several papers had been already published (and several of them are cited here). So, it should be discussed in the Introduction. And in the introduction the authors need to state more clearly what did hey want to study here. It is important to understand the novelty of the study.
Response: It has been documented that miRNAs are expressed in species and tissue specific manner. However, miR-153 (Wang et al. 2016) and miR-28 (Yang et al. 2011) studied before in different species and cell types (mainly cancer cell lines). These facts are mentioned in the introduction part (line: 78-80). However, the involvement of those miRNAs in Nrf2 regulation in bovine granulosa cells is still unclear.
Comment: Usage of H2O2 as a prooxidant: why did you choose to use 10 µM concentration (so low)?
Response: We have selected the 5µM H2O2 in the present experiment based on the dose dependent experiment performed and published before (Saeed-Zidane et al. 2017), where where different H2O2 concentrations (2.5, 5, 10, 20 and 50µM) were investigated. This concentration was the level in which we could induce ROS without toxic effect on cells.
Comment: Fig 1C and Fig. 9: quantification of Nrf2 levels should be also done at the protein level (for example, by Western blot), as Nrf2 is subjected to proteolysis by Keap1 and is stabilized upon dissociation. In addition, it should be explained why Nrf2 levels was accessed at mRNA level.
Response: The differential activity of NRF2 can only be determined by using ICC as it can revealed the cytoplasmic and nuclear NRF2 in cells. By ICC we could clearly indicate the active and dormant form of NRF2 as it has been indicated in figure 5 and 7. The main focus of the experiment in figure 1 and 9 is to show the effect of hydrogen peroxide and Querecetin on miRNA expression, respectively.
Comment: Mitotracker assay that was used to access mitochondrial activity. if I am not mistaken, this dye stains thiol groups in mitochondria. Thus, its staining correlates with mitochondria mass and not with its activity. So, it should be checked and revised if needed throughout the text (as it puts into question one of the conclusions). Moreover, I am not sure that the conclusion on Fig. 6A is correct, as H2O2 triggers oxidation of thiol groups. Thus, a decrease in mitotracker staining here may show not a decrease in mitochondria mass but merely oxidation of thiol groups. So, a conclusion needs to be verified by other method as well.
Response: MitoTracker® Red CMXRos was used for investigate the mitochondrial activity in granulosa cells. MitoTracker Red CMXRos is a red-fluorescent dye that stains mitochondria in live cells and its accumulation is dependent upon membrane potential (MMP), which is a marker for mitochondrial functionality (Sakamuru et al. 2016; Poot et al. 2017). Moreover, CMXRos exhibits good photostability and responds to changes in MMP and detecting functionally intact mitochondria. In other words, this probe allows for identification of active mitochondria. Recently CMXRos was used in several researches to measure mitochondrial function (Sorvina et al. 2018; Caicedo et al. 2015; Maharjan et al. 2014; Claus et al. 2011; Miettinen and Björklund 2016). That it the standard way of presenting the association between ROS and mitochondrial function.
Comment: Line 148: The conclusion about inadequate mitochondrial distribution is not supported by the figure, as the image was captured at low resolution, and the mitochondrial structure (form, network) is not seen at all. So, images at better resolution are required.
Response: Our aim here was to investigate the effect of oxidative stress on mitochondrial function instead of mitochondrial structure, form and network. That is why we changed the term mitochondrial distribution and replaced by mitochondrial function.
Comment: Measurement of ROS was performed with H2DCFDA. However, this dye is also oxidized by peroxynitrite, and sometimes its fluorescence depends on redistribution of iron from mitochondria. So, to meet currently required experimental level, these experiments need to be verified with other dyes (such as dihydroethidium or any other).
Response: This kit is the most widely used kit for detection of ROS in various experiments and cell types in response to intra- or extracellular activation with oxidative stimulus (Biosa et al. 2018; Egan et al. 2007; Oparka et al. 2016; Sypniewski et al. 2018; Yoon et al. 2018). So the usage of the kit DCFH-DA for intracellular ROS detection induced by H2O2 was justifiable and the result obtained using this kit is reliable enough.
Comment: Fig. 5B and 7B: H2O2 triggered elevation of Nrf2 level in the cells, However, basing on the figure it is impossible to say that the transcription factor changed its localization (translocated into the nucleus, as it should have done). And it is a bit strange, taking into account that in these cells there is a clear up-regulation of Nqo1 and HO-1. On Fig 7B it is quite clear that the majority of Nrf2 is outside the nuclei. So, the authors probably should consider analyzing Nrf2 localization (for example, by separating fractions of nuclear and cytoplasmic proteins using NE-PER kit with subsequent Nrf2 measurement by western blotting. It should strengthen the conclusion.
Response: The presentation of nuclear and cytoplasmic NRF2 using ICC is the only option we have and should justify the results we have. On top of that, regulation of the downstream target transcripts is in NRF2 dependent manner, which signifies the nuclear translocation and binding of NRF2 onto the ARE of the target transcripts. As the reviewer highlighted, analyzing the nuclear and cytoplasmic proteins separately could provide better understanding.
Comment: Fig. 10. There is an indication that Nrf2-responsive genes stimulate mitochondria (activity, mass??). But strictly speaking, the authors did not access any mechanisms (transcription factor for example) that are known to affect mitochondria biogenesis. As one, Nrf1 (nuclear respiratory factor 1) could be named. So, it could be worth mentioning this limitation in the discussion and pointing out to necessity for such studies in future.
Response: We have changed the arrow connecting NRF2 and mitochondria since the effect of NRF2 activity in mitochondrial function is by reducing the ROS accumulation as it can be seen a dashed line between reduced ROS and mitochondrial activity.
Minor issues
Comment: Please revise the text. For example, pay attention to the sentence on lines 58-64: it should be in present tense and in third form.
Response: We have revised the manuscript for English grammar errors .
Comment: Superoxide anion should be written correctly (O2●-), as it is a radical actually. So, revise it in the lines 43 and 249.
Response: Corrected as per the recommendation of the reviewer
Comment: Line 93: HMOX1 is a gene. If the authors wanted to name the corresponding protein, it should have been HO-1.
Response: Corrected as per the recommendation of the reviewer
Comment: Fig 2A, 4A, : please state that SOD was SOD1, and Prdx was Prdx1. By the way, why of several Prdx isoforms that are regulated by Nrf2, you chose Prdx1.
Response: Corrected as per the recommendation of the reviewer. We selected PRDX1 as a candidate gene regulated by Nrf2. Furthermore, PRDX1 also regulates several ROS‐dependent signaling pathways and is thought to be a key intracellular intermediate balancing cell survival and apoptosis (Neumann et al. 2009; Wood et al. 2003). Moreover, Compared with other antioxidant enzymes, PRDX1 employs a particular mechanism to detoxify peroxide with reducing equivalents provided through the thioredoxin (Trx) system (Zhang et al. 2009), and play a role in suppressing stress‐induced cell death through ROS‐dependent signalling pathway (Ding et al. 2017)
Comment: Discussion: In the introduction the authors discussed that Nrf2 controls expression of Nqo1, HO-1, Prdx1, CAT, and SOD1. But the experimental data here show absence of such control for the latter three genes. So it could be mentioned in the discussion.
Response: The activation of antioxidants by elevated NRF2 expression is very differential as it has been observed in our previous experiments (Saeed-Zidane et al. 2017).
Comment: Line 416: Please write in which buffer/media the cells were incubated with H2DCFDA, and what was the final concentration of the dye.
Response: It is indicated as recommended by reviewer.
Publication bibliography
Athale, Janhavi; Ulrich, Allison; MacGarvey, Nancy Chou; Bartz, Raquel R.; Welty-Wolf, Karen E.; Suliman, Hagir B.; Piantadosi, Claude A. (2012): Nrf2 promotes alveolar mitochondrial biogenesis and resolution of lung injury in Staphylococcus aureus pneumonia in mice. In Free radical biology & medicine 53 (8), pp. 1584–1594. DOI: 10.1016/j.freeradbiomed.2012.08.009.
Biosa, Alice; Sanchez-Martinez, Alvaro; Filograna, Roberta; Terriente-Felix, Ana; Alam, Sarah M.; Beltramini, Mariano et al. (2018): Superoxide dismutating molecules rescue the toxic effects of PINK1 and parkin loss. In Human molecular genetics 27 (9), pp. 1618–1629. DOI: 10.1093/hmg/ddy069.
Caicedo, Andrés; Fritz, Vanessa; Brondello, Jean-Marc; Ayala, Mickaël; Dennemont, Indira; Abdellaoui, Naoill et al. (2015): MitoCeption as a new tool to assess the effects of mesenchymal stem/stromal cell mitochondria on cancer cell metabolism and function. In Scientific Reports 5, 9073 EP -. DOI: 10.1038/srep09073.
Claus, C.; Chey, S.; Heinrich, S.; Reins, M.; Richardt, B.; Pinkert, S. et al. (2011): Involvement of p32 and microtubules in alteration of mitochondrial functions by rubella virus. In Journal of virology 85 (8), pp. 3881–3892. DOI: 10.1128/JVI.02492-10.
Ding, Chenbo; Fan, Xiaobo; Wu, Guoqiu (2017): Peroxiredoxin 1 - an antioxidant enzyme in cancer. In Journal of cellular and molecular medicine 21 (1), pp. 193–202. DOI: 10.1111/jcmm.12955.
Egan, Martin J.; Wang, Zheng-Yi; Jones, Mark A.; Smirnoff, Nicholas; Talbot, Nicholas J. (2007): Generation of reactive oxygen species by fungal NADPH oxidases is required for rice blast disease. In Proceedings of the National Academy of Sciences of the United States of America 104 (28), pp. 11772–11777. DOI: 10.1073/pnas.0700574104.
Holmström, Kira M.; Kostov, Rumen V.; Dinkova-Kostova, Albena T. (2016): The multifaceted role of Nrf2 in mitochondrial function. In Current opinion in toxicology 1, pp. 80–91. DOI: 10.1016/j.cotox.2016.10.002.
Maharjan, Sunita; Oku, Masahide; Tsuda, Masashi; Hoseki, Jun; Sakai, Yasuyoshi (2014): Mitochondrial impairment triggers cytosolic oxidative stress and cell death following proteasome inhibition. In Scientific Reports 4, 5896 EP -. DOI: 10.1038/srep05896.
Miettinen, Teemu P.; Björklund, Mikael (2016): Cellular Allometry of Mitochondrial Functionality Establishes the Optimal Cell Size. In Developmental cell 39 (3), pp. 370–382. DOI: 10.1016/j.devcel.2016.09.004.
Neumann, Carola A.; Cao, Juxiang; Manevich, Yefim (2009): Peroxiredoxin 1 and its role in cell signaling. In Cell cycle (Georgetown, Tex.) 8 (24), pp. 4072–4078. DOI: 10.4161/cc.8.24.10242.
Oparka, Monika; Walczak, Jarosław; Malinska, Dominika; van Oppen, Lisanne M. P. E.; Szczepanowska, Joanna; Koopman, Werner J. H.; Wieckowski, Mariusz R. (2016): Quantifying ROS levels using CM-H2DCFDA and HyPer. In Methods (San Diego, Calif.) 109, pp. 3–11. DOI: 10.1016/j.ymeth.2016.06.008.
Poot, M.; Zhang, Y. Z.; Krämer, J. A.; Wells, K. S.; Jones, L. J.; Hanzel, D. K. et al. (2017): Analysis of mitochondrial morphology and function with novel fixable fluorescent stains. In J Histochem Cytochem. 44 (12), pp. 1363–1372. DOI: 10.1177/44.12.8985128.
Reddy, Narsa M.; Qureshi, Wajiha; Potteti, Haranath; Kalvakolanu, Dhananjaya V.; Reddy, Sekhar P. (2014): Regulation of Mitochondrial Functions by Transcription Factor NRF2. In Viswanathan Natarajan, Narasimham L. Parinandi (Eds.): Mitochondrial Function in Lung Health and Disease, vol. 15. New York, NY: Springer New York (Respiratory Medicine), pp. 27–50.
Saeed-Zidane, Mohammed; Linden, Lea; Salilew-Wondim, Dessie; Held, Eva; Neuhoff, Christiane; Tholen, Ernst et al. (2017): Cellular and exosome mediated molecular defense mechanism in bovine granulosa cells exposed to oxidative stress. In PloS one 12 (11), e0187569. DOI: 10.1371/journal.pone.0187569.
Sakamuru, Srilatha; Attene-Ramos, Matias S.; Xia, Menghang (2016): Mitochondrial Membrane Potential Assay. In Methods in molecular biology (Clifton, N.J.) 1473, pp. 17–22. DOI: 10.1007/978-1-4939-6346-1_2.
Sorvina, Alexandra; Bader, Christie A.; Darby, Jack R. T.; Lock, Mitchell C.; Soo, Jia Yin; Johnson, Ian R. D. et al. (2018): Mitochondrial imaging in live or fixed tissues using a luminescent iridium complex. In Scientific Reports 8 (1), p. 8191. DOI: 10.1038/s41598-018-24672-w.
Sypniewski, Daniel; Szkaradek, Natalia; Loch, Tomasz; Waszkielewicz, Anna M.; Gunia-Krzyżak, Agnieszka; Matczyńska, Daria et al. (2018): Contribution of reactive oxygen species to the anticancer activity of aminoalkanol derivatives of xanthone. In Investigational new drugs 36 (3), pp. 355–369. DOI: 10.1007/s10637-017-0537-x.
Wang, Bo; Teng, Yang; Liu, Qilun (2016): MicroRNA-153 Regulates NRF2 Expression and is Associated with Breast Carcinogenesis. In Clinical laboratory 62 (1-2), pp. 39–47.
Wood, Zachary A.; Schröder, Ewald; Robin Harris, J.; Poole, Leslie B. (2003): Structure, mechanism and regulation of peroxiredoxins. In Trends in biochemical sciences 28 (1), pp. 32–40.
Yang, Muhua; Yao, Yuan; Eades, Gabriel; Zhang, Yongshu; Zhou, Qun (2011): MiR-28 regulates Nrf2 expression through a Keap1-independent mechanism. In Breast cancer research and treatment 129 (3), pp. 983–991. DOI: 10.1007/s10549-011-1604-1.
Yoon, Dong Suk; Lee, Myon-Hee; Cha, Dong Seok (2018): Measurement of Intracellular ROS in Caenorhabditis elegans Using 2',7'-Dichlorodihydrofluorescein Diacetate. In Bio-protocol 8 (6). DOI: 10.21769/BioProtoc.2774.
Zhang, Bo; Wang, Yan; Su, Yongping (2009): Peroxiredoxins, a novel target in cancer radiotherapy. In Cancer letters 286 (2), pp. 154–160. DOI: 10.1016/j.canlet.2009.04.043.
Reviewer 2 Report
Significant work done by Dr. Tesfaye. This work signifies the role of NRF2 in bovine granulosa cells. Critical work which supports hypothesis of the work.
The main theme of this work or main question of this study is to determine the role of noncoding mRNAs as modulator of NRF2 pathways including investigating the role of Quercetin as a regulator of NRF2 signaling.
The question has a translational aspect in case of understanding the complex role of NRF2 pathway for ovarian function which is very relevant to current understanding of NRF2's modulation.
Even though there are several works done on this line, this study tries to answer some of the unresolved concerns. For example, the role of miRNAs in NRF2 regulation in bovine granulosa cells were not clearly understood. In this study they have tried to resolve that concern. Here they have concentrated on 3 candidates, specifically miR-153, 708 and 28's modulating roles of NRF2 signaling in bovine granulosa cells. In addition, they also tried to investigate the role Que as exogeneous modulator on NRF2 regulation in bovine granulosa cells, which was not completely analyzed previously.
Paper is well written and easy to understand. Some minor corrections can be formatted during proof reading.
To my understanding there is enough evidence which clearly supports the hypothesis of the work. Authors thoroughly made scientific arguments to support their work and which I think clearly convincing to address the main question of this work.
Please feel free to enquire more if needed. But am confident enough on this manuscript for the acceptance in IJMS.
Author Response
We sincerely appreciate the reviewers for the valuable comments
Reviewer #2:
Comment: Significant work done by Dr. Tesfaye. This work signifies the role of NRF2 in bovine granulosa cells. Critical work which supports hypothesis of the work.
The main theme of this work or main question of this study is to determine the role of noncoding mRNAs as modulator of NRF2 pathways including investigating the role of Quercetin as a regulator of NRF2 signaling.
The question has a translational aspect in case of understanding the complex role of NRF2 pathway for ovarian function, which is very relevant to current understanding of NRF2's modulation.
Even though there are several works done on this line, this study tries to answer some of the unresolved concerns. For example, the role of miRNAs in NRF2 regulation in bovine granulosa cells were not clearly understood. In this study they have tried to resolve that concern. Here they have concentrated on 3 candidates, specifically miR-153, 708 and 28's modulating roles of NRF2 signaling in bovine granulosa cells. In addition, they also tried to investigate the role Que as exogeneous modulator on NRF2 regulation in bovine granulosa cells, which was not completely analyzed previously.
Paper is well written and easy to understand. Some minor corrections can be formatted during proof reading.
To my understanding there is enough evidence which clearly supports the hypothesis of the work. Authors thoroughly made scientific arguments to support their work and which I think clearly convincing to address the main question of this work.Please feel free to enquire more if needed. But am confident enough on this manuscript for the acceptance in IJMS.
Response: Thanks for the constructive comments made. We have done our best to make the revised manuscript better than before by revising it in all aspects.
Round 2
Reviewer 1 Report
The authors have revised the manuscript taking onto account all minor critique and some of the major concerns. For the other questions they presented a detailed response. However, in my personal opinion several questions still remain unresolved (Q1, Q6, and Q7).
Q1: One of the major questions to this study was the fact that two out of three miRNAs studied here were already reported to regulate Nrf2 expression. This raised the question about novelty of the present study. During revisions the authors merely noted the fact in the Introduction providing one reference per each miRNA but stated that it was shown not in the cells they study, and so this question should have been studied more extensively. For miR153 the authors cited the paper of Wang et al (2016) but omitted the ones from Narasimhan et al (2012, 2014), Liu et al (2017), and Zhu et al (2018). Existence of so many papers on this particular miRNA in my personal opinion requires discussion of a place of the current findings in the context of literature data.
Q6: Insufficient usage of DCFHDA as a single dye for ROS measurement. The authors answered that this dye is the most commonly used in the world thus justifying its usage as a single method. In support, the authors provided a list of five references where the authors did use this dye. However, it is widely accepted that DCF fluorescent is also affected by reactive nitrogen and carbon species and can be affected by altered iron metabolism (see a consensus-like paper doi: 10.1016/j.freeradbiomed.2011.09.030 - with a conclusion that "recommendation to researchers in this field is that the DCFH-DA probe cannot be reliably used to measure intracellular H2O2 and other reactive oxygen species."). Noteworthy also that in two out of five cited by the authors papers other techniques for ROS production assessment had been used: peroxide-selective HyPER proteins in a study of Oparka et al and superoxide-selective NBD probe by Egan et al. So it is difficult to understand why the authors are reluctant to use other dyes that can ensure that the signal is indeed due to enhanced ROS production and provide evidence that ROS are accumulated in mitochondria for instance (i.e. in case of MitoSOX usage).
In addition here, it is not clear from Experimental section what was the final concentration of the dye.
Q7: The authors admitted that ICC is the only available for them method for assessment of Nrf2 status which is normal. But here the authors showed low-resolution images while in their previous study (Sulforaphane protects granulosa cells against oxidative stress via activation of NRF2-ARE pathway) they published perfect images showing very clear translocation of Nrf2 from cytoplasm into nucleus. Why? But yes, the conclusions on Fig 5 are supported by activation of Nqo1 (Nrf2-dependent gene) expression.
Author Response
We sincerely appreciate the reviewers for the valuable comments
Reviewer #1:
Comment: One of the major questions to this study was the fact that two out of three miRNAs studied here were already reported to regulate Nrf2 expression. This raised the question about novelty of the present study. During revisions the authors merely noted the fact in the Introduction providing one reference per each miRNA but stated that it was shown not in the cells they study, and so this question should have been studied more extensively. For miR153 the authors cited the paper of Wang et al (2016) but omitted the ones from Narasimhan et al (2012, 2014), Liu et al (2017), and Zhu et al (2018). Existence of so many papers on this particular miRNA in my personal opinion requires discussion of a place of the current findings in the context of literature data.
Response: As per the recommendation of reviewer´s comment previous studies on those candidate miRNAs in various cells types and experimental setup.
Comment: Insufficient usage of DCFHDA as a single dye for ROS measurement. The authors answered that this dye is the most commonly used in the world thus justifying its usage as a single method. In support, the authors provided a list of five references where the authors did use this dye. However, it is widely accepted that DCF fluorescent is also affected by reactive nitrogen and carbon species and can be affected by altered iron metabolism (see a consensus-like paper doi: 10.1016/j.freeradbiomed.2011.09.030 - with a conclusion that "recommendation to researchers in this field is that the DCFH-DA probe cannot be reliably used to measure intracellular H2O2 and other reactive oxygen species."). Noteworthy also that in two out of five cited by the authors papers other techniques for ROS production assessment had been used: peroxide-selective HyPER proteins in a study of Oparka et al and superoxide-selective NBD probe by Egan et al. So it is difficult to understand why the authors are reluctant to use other dyes that can ensure that the signal is indeed due to enhanced ROS production and provide evidence that ROS are accumulated in mitochondria for instance (i.e. in case of MitoSOX usage).
Oparka et al and Egan et al used HyPER proteins and NBD probe for selective measurement of peroxide and superoxide, respectively. However, both of them used DCFH-DA probe for the detection of general and other oxidant levels.
Regarding to the final concentration of the dye, it was mentioned in (line 443)
Response: The primary purpose of detecting ROS level in the present study is just to evidence the induction of ROS accumulation due to Hydrogen peroxide treatment. Detection of ROS exclusively in connection with mitochondria was not the aim of our study. Therefore, the usage of the DCFH-DA kit for intracellular ROS detection induced by H2O2 treatment was justifiable and the result obtained using this kit is reliable enough to meet the purpose of the study.
Comment: The authors admitted that ICC is the only available for them method for assessment of Nrf2 status which is normal. But here the authors showed low-resolution images while in their previous study (Sulforaphane protects granulosa cells against oxidative stress via activation of NRF2-ARE pathway) they published perfect images showing very clear translocation of Nrf2 from cytoplasm into nucleus. Why? But yes, the conclusions on Fig 5 are supported by activation of Nqo1 (Nrf2-dependent gene) expression.
Response: As indicated by the reviewer the best way of showing the distribution of NRF2 protein in cultured cells as we have also shown in our previous paper (Saeed-Zidan et al. 2018) is ICC. As per our observation ICC data presented in this manuscript is much more clear than the one we have presented previously (Sulforaphane protects granulosa cells against oxidative stress via activation of NRF2-ARE pathway). The clarity of the results are more evident when one looks into the merged figures of NRF2 protein and DAPI staining. As it has also been reported by our previous papers the activation of NRF2 protein resulted in differential activation of antioxidants in various experimental setups.